# Adding One Neuron Can Eliminate All Bad Local Minima

**Shiyu Liang**
Coordinated Science Laboratory
Dept. of Electrical and Computer Engineering
University of Illinois at Urbana-Champaign
sliang26@illinois.edu

**Ruoyu Sun**
Coordinated Science Laboratory
Department of ISE
University of Illinois at Urbana-Champaign
ruoyus@illinois.edu

**Jason D. Lee**
Marshall School of Business
University of Southern California
jasonlee@marshall.usc.edu

**R. Srikant**
Coordinated Science Laboratory
Dept. of Electrical and Computer Engineering
University of Illinois at Urbana-Champaign
rsrikant@illinois.edu
*

## Abstract

One of the main difficulties in analyzing neural networks is the non-convexity of the loss function which may have many bad local minima. In this paper, we study the landscape of neural networks for binary classification tasks. Under mild assumptions, we prove that after adding one special neuron with a skip connection to the output, or one special neuron per layer, every local minimum is a global minimum.

## 1 Introduction

Deep neural networks have recently achieved huge success in various machine learning tasks (see, Krizhevsky et al. 2012; Goodfellow et al. 2013; Wan et al. 2013, for example). However, a theoretical understanding of neural networks is largely lacking. One of the difficulties in analyzing neural networks is the non-convexity of the loss function which allows the existence of many local minima with large losses. This was long considered a bottleneck of neural networks, and one of the reasons why convex formulations such as support vector machine (Cortes & Vapnik, 1995) were preferred previously. Given the recent empirical success of the deep neural networks, an interesting question is whether the non-convexity of the neural network is really an issue.

It has been widely conjectured that all local minima of the empirical loss lead to similar training performance (LeCun et al., 2015; Choromanska et al., 2015). For example, prior works empirically showed that neural networks with identical architectures but different initialization points can converge to local minima with similar classification performance (Krizhevsky et al., 2012; He et al., 2016; Huang & Liu, 2017). On the theoretical side, there have been many recent attempts to analyze the landscape of the neural network loss functions. A few works have studied deep networks, but they either require linear activation functions (Baldi & Hornik, 1989; Kawaguchi, 2016; Freeman & Bruna, 2016; Hardt & Ma, 2017; Yun et al., 2017), or require assumptions such as independence of ReLU activations (Choromanska et al., 2015) and significant overparametrization (Nguyen & Hein, 2017a,b; Livni et al., 2014). There is a large body of works that study single-hidden-layer neural networks and provide various conditions under which a local search algorithm can find a global minimum (Du & Lee, 2018; Ge et al., 2018; Andoni et al., 2014; Sedghi & Anandkumar, 2014; Janzamin et al., 2015; Haeffele & Vidal, 2015; Gautier et al., 2016; Brutzkus & Globerson,

2017; Soltanolkotabi, 2017; Soudry & Hoffer, 2017; Goel & Klivans, 2017; Du et al., 2017; Zhong et al., 2017; Li & Yuan, 2017; Liang et al., 2018; Mei et al., 2018). It can be roughly divided into two categories: non-global landscape analysis and global landscape analysis. For the first category, the result do not apply to all local minima. One typical conclusion is about the local geometry, i.e., in a small neighborhood of the global minima no bad local minima exist (Zhong et al., 2017; Du et al., 2017; Li & Yuan, 2017). Another typical conclusion is that a subset of local minima are global minima (Haeffele et al., 2014; Haeffele & Vidal, 2015; Soudry & Carmon, 2016; Nguyen & Hein, 2017a,b). Shamir (2018) has shown that a subset of second-order local minima can perform nearly as well as linear predictors. The presence of various conclusions reflects the difficulty of the problem: while analyzing the global landscape seems hard, we may step back and analyze the local landscape or a "majority" of the landscape. For the second category of global landscape analysis, the typical result is that every local minimum is a global minimum. However, even for single-layer networks, strong assumptions such as over-parameterization, very special neuron activation functions, fixed second layer parameters and/or Gaussian data distribution are often needed in the existing works. The presence of various strong assumptions also reflects the difficulty of the problem: even for the single-hidden-layer nonlinear neural network, it seems hard to analyze the landscape, so it is reasonable to make various assumptions.

One exception is the recent work Liang et al. (2018) which adopts a different path: instead of simply making several assumptions to obtain positive results, it carefully studies the effect of various conditions on the landscape of neural networks for binary classification. It gives both positive and negative results on the existence of bad local minimum under different conditions. In particular, it studies many common types of neuron activation functions and shows that for a class of neurons there is no bad local minimum, and for other neurons there is. This clearly shows that the choice of neurons can affect the landscape. Then a natural question is: while Liang et al. (2018) considers some special types of data and a broad class of neurons, can we obtain results for more general data when limiting to a smaller class of neurons?

## 1.1 Our Contributions

Given this context, our main result is quite surprising: for a neural network with a special type of neurons, every local minimum is a global minimum of the loss function. Our result requires no assumption on the network size, the specific type of the original neural network, etc., yet our result applies to every local minimum. Besides the requirement on the neuron activation type, the major trick is an associated regularizer. Our major results and their implications are as follows:

- We focus on the binary classification problem with a smooth hinge loss function. We prove the following result: for any neural network, by adding a special neuron (e.g., exponential neuron) to the network and adding a quadratic regularizer of this neuron, the new loss function has no bad local minimum. In addition, every local minimum achieves the minimum misclassification error.

- In the main result, the augmented neuron can be viewed as a skip connection from the input to the output layer. However, this skip connection is not critical, as the same result also holds if we add one special neuron to each layer of a fully-connected feedforward neural network.

- To our knowledge, this is the first result that no spurious local minimum exists for a wide class of deep nonlinear networks. Our result indicates that the class of "good neural networks" (neural networks such that there is an associated loss function with no spurious local minima) contains any network with one special neuron, thus this class is rather "dense" in the class of all neural networks: the distance between any neural network and a good neural network is just a neuron away.

The outline of the paper is as follows. In Section 2, we present several notations. In Section 3, we present the main result and several extensions on the main results are presented in Section 4. We present the proof idea of the main result in Section 5 and conclude this paper in Section 6. All proofs are presented in Appendix.

## 2 Preliminaries

**Feed-forward networks.** Given an input vector of dimension $d$, we consider a neural network with $L$ layers of neurons for binary classification. We denote by $M_l$ the number of neurons in the $l$-th layer (note that $M_0 = d$). We denote the neural activation function by $\sigma$. Let $\boldsymbol{W}_l \in \mathbb{R}^{M_{l-1} \times M_l}$ denote the weight matrix connecting the $(l-1)$-th and $l$-th layer and $\boldsymbol{b}_l$ denote the bias vector for neurons in

the $l$-th layer. Let $\boldsymbol{W}_{L+1} \in \mathbb{R}^{M_L}$ and $b_L \in \mathbb{R}$ denote the weight vector and bias scalar in the output layer, respectively. Therefore, the output of the network $f : \mathbb{R}^d \to \mathbb{R}$ can be expressed by

$$f(x;\boldsymbol{\theta}) = \boldsymbol{W}_{L+1}^{\top} \boldsymbol{\sigma} \left( \boldsymbol{W}_L \boldsymbol{\sigma} \left( ... \boldsymbol{\sigma} \left( \boldsymbol{W}_1^{\top} x + \boldsymbol{b}_1 \right) + \boldsymbol{b}_{L-1} \right) + \boldsymbol{b}_L \right) + b_{L+1}. \tag{1}$$

**Loss and error**. We use $\mathcal{D} = \{(x_i, y_i)\}_{i=1}^{n}$ to denote a dataset containing $n$ samples, where $x_i \in \mathbb{R}^d$ and $y_i \in \{-1, 1\}$ denote the feature vector and the label of the $i$-th sample, respectively. Given a neural network $f(x;\boldsymbol{\theta})$ parameterized by $\boldsymbol{\theta}$ and a loss function $\ell : \mathbb{R} \to \mathbb{R}$, in binary classification tasks, we define the empirical loss $L_n(\boldsymbol{\theta})$ as the average loss of the network $f$ on a sample in the dataset and define the training error (also called the misclassification error) $R_n(\boldsymbol{\theta}; f)$ as the misclassification rate of the network $f$ on the dataset $\mathcal{D}$, i.e.,

$$L_n(\boldsymbol{\theta}) = \sum_{i=1}^{n} \ell(-y_i f(x_i;\boldsymbol{\theta})) \quad \text{and} \quad R_n(\boldsymbol{\theta}; f) = \frac{1}{n} \sum_{i=1}^{n} \mathbb{I}\{y_i \neq \mathrm{sgn}(f(x_i;\boldsymbol{\theta}))\}. \tag{2}$$

where $\mathbb{I}$ is the indicator function.

**Tensors products.** We use $\boldsymbol{a} \otimes \boldsymbol{b}$ to denote the tensor product of vectors $\boldsymbol{a}$ and $\boldsymbol{b}$ and use $\boldsymbol{a}^{\otimes k}$ to denote the tensor product $\boldsymbol{a} \otimes ... \otimes \boldsymbol{a}$ where $\boldsymbol{a}$ appears $k$ times. For an $N$-th order tensor $\boldsymbol{T} \in \mathbb{R}^{d_1 \times d_2 \times ... \times d_N}$ and $N$ vectors $\boldsymbol{u}_1 \in \mathbb{R}^{d_1}, \boldsymbol{u}_2 \in \mathbb{R}^{d_2}, ..., \boldsymbol{u}_N \in \mathbb{R}^{d_N}$, we define

$$\boldsymbol{T} \otimes \boldsymbol{u}_1 ... \otimes \boldsymbol{u}_N = \sum_{i_1 \in [d_1], ..., i_N \in [d_N]} \boldsymbol{T}(i_1, ..., i_N) \boldsymbol{u}_1(i_1) ... \boldsymbol{u}_N(i_N),$$

where we use $\boldsymbol{T}(i_1, ..., i_N)$ to denote the $(i_1, ..., i_N)$-th component of the tensor $\boldsymbol{T}$, $\boldsymbol{u}_k(i_k)$ to denote the $i_k$-th component of the vector $\boldsymbol{u}_k$, $k = 1, ..., N$ and $[d_k]$ to denote the set $\{1, ..., d_k\}$.

## 3 Main Result

In this section, we first present several important conditions on the loss function and the dataset in order to derive the main results. After that, we will present the main results.

### 3.1 Assumptions

In this subsection, we introduce two assumptions on the loss function and the dataset.

**Assumption 1 (Loss function)** *Assume that the loss function $\ell : \mathbb{R} \to \mathbb{R}$ is monotonically non-decreasing and twice differentiable, i.e., $\ell \in C^2$. Assume that every critical point of the loss function $\ell(z)$ is also a global minimum and every global minimum $z$ satisfies $z < 0$.*

A simple example of the loss function satisfying Assumption 1 is the polynomial hinge loss, i.e., $\ell(z) = [\max\{z+1, 0\}]^p, p \geq 3$. It is always zero for $z \leq -1$ and behaves like a polynomial function in the region $z > -1$. Note that the condition that every global minimum of the loss function $\ell(z)$ is negative is not needed to prove the result that every local minimum of the empirical loss is globally minimal, but is necessary to prove that the global minimizer of the empirical loss is also the minimizer of the misclassification rate.

**Assumption 2 (Realizability)** *Assume that there exists a set of parameters $\boldsymbol{\theta}$ such that the neural network $f(\cdot;\boldsymbol{\theta})$ is able to correctly classify all samples in the dataset $\mathcal{D}$.*

By Assumption 2, we assume that the dataset is realizable by the neural architecture $f$. We note that this assumption is consistent with previous empirical observations (Zhang et al., 2016; Krizhevsky et al., 2012; He et al., 2016) showing that at the end of the training process, neural networks usually achieve zero misclassification rates on the training sets. However, as we will show later, if the loss function $\ell$ is convex, then we can prove the main result even without Assumption 2.

### 3.2 Main Result

In this subsection, we first introduce several notations and next present the main result of the paper. Given a neural architecture $f(\cdot;\boldsymbol{\theta})$ defined on a $d$-dimensional Euclidean space and parameterized by a set of parameters $\boldsymbol{\theta}$, we define a new architecture $\tilde{f}$ by adding the output of an exponential neuron to the output of the network $f$, i.e.,

$$\tilde{f}(x, \tilde{\boldsymbol{\theta}}) = f(x;\boldsymbol{\theta}) + a \exp\left(\boldsymbol{w}^{\top} x + b\right), \tag{3}$$

where the vector $\tilde{\boldsymbol{\theta}} = (\boldsymbol{\theta}, a, \boldsymbol{w}, b)$ denote the parametrization of the network $\tilde{f}$. For this designed model, we define the empirical loss function as follows,

$$\tilde{L}_n(\tilde{\boldsymbol{\theta}}) = \sum_{i=1}^{n} \ell\left(-y_i \tilde{f}(x; \tilde{\boldsymbol{\theta}})\right) + \frac{\lambda a^2}{2}, \tag{4}$$

where the scalar $\lambda$ is a positive real number, i.e., $\lambda > 0$. Different from the empirical loss function $L_n$, the loss $\tilde{L}_n$ has an additional regularizer on the parameter $a$, since we aim to eliminate the impact of the exponential neuron on the output of the network $\tilde{f}$ at every local minimum of $\tilde{L}_n$. As we will show later, the exponential neuron is inactive at every local minimum of the empirical loss $\tilde{L}_n$. Now we present the following theorem to show that every local minimum of the loss function $\tilde{L}_n$ is also a global minimum.

**Remark:** Instead of viewing the exponential term in Equation (3) as a neuron, one can also equivalently think of modifying the loss function to be

$$\tilde{L}_n(\tilde{\boldsymbol{\theta}}) = \sum_{i=1}^{n} \ell\left(-y_i(f(x_i; \boldsymbol{\theta}) + a \exp(\boldsymbol{w}^\top x_i + b))\right) + \frac{\lambda a^2}{2}.$$

Then, one can interpret Equation (3) and (4) as maintaining the original neural architecture and slightly modifying the loss function.

**Theorem 1** *Suppose that Assumption 1 and 2 hold. Then both of the following statements are true:*

(i) *The empirical loss function $\tilde{L}_n(\tilde{\boldsymbol{\theta}})$ has at least one local minimum.*

(ii) *Assume that $\tilde{\boldsymbol{\theta}}^* = (\boldsymbol{\theta}^*, a^*, \boldsymbol{w}^*, b^*)$ is a local minimum of the empirical loss function $\tilde{L}_n(\tilde{\boldsymbol{\theta}})$, then $\tilde{\boldsymbol{\theta}}^*$ is a global minimum of $\tilde{L}_n(\tilde{\boldsymbol{\theta}})$. Furthermore, $\boldsymbol{\theta}^*$ achieves the minimum loss value and the minimum misclassification rate on the dataset $\mathcal{D}$, i.e., $\boldsymbol{\theta}^* \in \arg\min_{\boldsymbol{\theta}} L_n(\boldsymbol{\theta})$ and $\boldsymbol{\theta}^* \in \arg\min_{\boldsymbol{\theta}} R_n(\boldsymbol{\theta}; f)$.*

**Remarks:** (i) Theorem 1 shows that every local minimum $\tilde{\boldsymbol{\theta}}^*$ of the empirical loss $\tilde{L}_n$ is also a global minimum and shows that $\boldsymbol{\theta}^*$ achieves the minimum training error and the minimum loss value on the original loss function $L_n$ at the same time. (ii) Since we do not require the explicit form of the neural architecture $f$, Theorem 1 applies to the neural architectures widely used in practice such as convolutional neural network (Krizhevsky et al., 2012), deep residual networks (He et al., 2016), etc. This further indicates that the result holds for any real neural activation functions such as rectified linear unit (ReLU), leaky rectified linear unit (Leaky ReLU), etc. (iii) As we will show in the following corollary, at every local minimum $\tilde{\boldsymbol{\theta}}^*$, the exponential neuron is inactive. Therefore, at every local minimum $\tilde{\boldsymbol{\theta}}^* = (\boldsymbol{\theta}^*, a^*, \boldsymbol{w}^*, b^*)$, the neural network $\tilde{f}$ with an augmented exponential neuron is equivalent to the original neural network $f$.

**Corollary 1** *Under the conditions of Theorem 1, if $\tilde{\boldsymbol{\theta}}^* = (\boldsymbol{\theta}^*, a^*, \boldsymbol{w}^*, b^*)$ is a local minimum of the empirical loss function $\tilde{L}_n(\tilde{\boldsymbol{\theta}})$, then two neural networks $f(\cdot; \boldsymbol{\theta}^*)$ and $\tilde{f}(\cdot; \tilde{\boldsymbol{\theta}}^*)$ are equivalent, i.e., $f(x; \boldsymbol{\theta}^*) = \tilde{f}(x; \tilde{\boldsymbol{\theta}}^*), \forall x \in \mathbb{R}^d$.*

Corollary 1 shows that at every local minimum, the exponential neuron does not contribute to the output of the neural network $\tilde{f}$. However, this does not imply that the exponential neuron is unnecessary, since several previous results (Safran & Shamir, 2018; Liang et al., 2018) have already shown that the loss surface of pure ReLU neural networks are guaranteed to have bad local minima. Furthermore, to prove the main result under any dataset, the regularizer is also necessary, since Liang et al. (2018) has already shown that even with an augmented exponential neuron, the empirical loss without the regularizer still have bad local minima under some datasets.

## 4 Extensions

### 4.1 Eliminating the Skip Connection

As noted in the previous section, the exponential term in Equation (3) can be viewed as a skip connection or a modification to the loss function. Our analysis also works under other architectures as well. When the exponential term is viewed as a skip connection, the network architecture is as shown in Fig. 1(a). This architecture is different from the canonical feedforward neural architectures

as there is a direct path from the input layer to the output layer. In this subsection, we will show that the main result still holds if the model $\tilde{f}$ is defined as a feedforward neural network shown in Fig. 1(b), where each layer of the network $f$ is augmented by an additional exponential neuron. This is a standard fully connected neural network except for one special neuron at each layer.

**Notations.** Given a fully-connected feedforward neural network $f(\cdot; \boldsymbol{\theta})$ defined by Equation (1), we define a new fully connected feedforward neural network $\tilde{f}$ by adding an additional exponential neuron to each layer of the network $f$. We use the vector $\tilde{\boldsymbol{\theta}} = (\boldsymbol{\theta}, \boldsymbol{\theta}_{\exp})$ to denote the parameterization of the network $\tilde{f}$, where $\boldsymbol{\theta}_{\exp}$ denotes the vector consisting of all augmented weights and biases. Let $\tilde{\boldsymbol{W}}_l \in \mathbb{R}^{(M_{l-1}+1)\times(M_l+1)}$ and $\tilde{\boldsymbol{b}}_l \in \mathbb{R}^{M_l+1}$ denote the weight matrix and the bias vector in the $l$-th layer of the network $\tilde{f}$, respectively. Let $\tilde{\boldsymbol{W}}_{L+1} \in \mathbb{R}^{(M_L+1)}$ and $\tilde{b}_{L+1} \in \mathbb{R}$ denote the weight vector and the bias scalar in the output layer of the network $\tilde{f}$, respectively. Without the loss of generality, we assume that the $(M_l+1)$-th neuron in the $l$-th layer is the augmented exponential neuron. Thus, the output of the network $\tilde{f}$ is expressed by

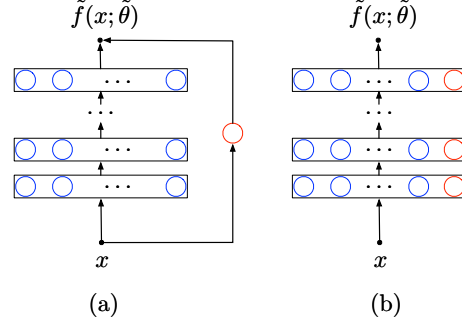

Figure 1: (a) The neural architecture considered in Theorem 1. (b) The neural architecture considered in Theorem 2. The blue and red circles denote the neurons $\sigma$ in the original network and the augmented exponential neurons, respectively.

$$\tilde{f}(x; \boldsymbol{\theta}) = \tilde{\boldsymbol{W}}_{L+1}^{\top} \tilde{\boldsymbol{\sigma}}_{L+1}\left(\tilde{\boldsymbol{W}}_L \tilde{\boldsymbol{\sigma}}_L\left(...\tilde{\boldsymbol{\sigma}}_1\left(\tilde{\boldsymbol{W}}_1^{\top} x + \tilde{\boldsymbol{b}}_1\right) + \tilde{\boldsymbol{b}}_{L-1}\right) + \tilde{\boldsymbol{b}}_L\right) + \tilde{b}_{L+1}, \qquad (5)$$

where $\tilde{\boldsymbol{\sigma}}_l : \mathbb{R}^{M_{l-1}+1} \to \mathbb{R}^{M_l+1}$ is a vector-valued activation function with the first $M_l$ components being the activation functions $\sigma$ in the network $f$ and with the last component being the exponential function, i.e., $\tilde{\boldsymbol{\sigma}}_l(z) = (\sigma(z), ..., \sigma(z), \exp(z))$. Furthermore, we use the $\tilde{\boldsymbol{w}}_l$ to denote the vector in the $(M_{l-1}+1)$-th row of the matrix $\tilde{\boldsymbol{W}}_l$. In other words, the components of the vector $\tilde{\boldsymbol{w}}_l$ are the weights on the edges connecting the exponential neuron in the $(l-1)$-th layer and the neurons in the $l$-th layer. For this feedforward network, we define an empirical loss function as

$$\tilde{L}_n(\tilde{\boldsymbol{\theta}}) = \sum_{i=1}^{n} \ell(-y_i \tilde{f}(x_i; \tilde{\boldsymbol{\theta}})) + \frac{\lambda}{2} \sum_{l=2}^{L+1} \|\tilde{\boldsymbol{w}}_l\|_{2L}^{2L} \qquad (6)$$

where $\|\boldsymbol{a}\|_p$ denotes the $p$-norm of a vector $\boldsymbol{a}$ and $\lambda$ is a positive real number, i.e., $\lambda > 0$. Similar to the empirical loss discussed in the previous section, we add a regularizer to eliminate the impacts of all exponential neurons on the output of the network. Similarly, we can prove that at every local minimum of $\tilde{L}_n$, all exponential neurons are inactive. Now we present the following theorem to show that if the set of parameters $\tilde{\boldsymbol{\theta}}^* = (\boldsymbol{\theta}^*, \boldsymbol{\theta}_{\exp}^*)$ is a local minimum of the empirical loss function $\tilde{L}_n(\tilde{\boldsymbol{\theta}})$, then $\tilde{\boldsymbol{\theta}}^*$ is a global minimum and $\boldsymbol{\theta}^*$ is a global minimum of both minimization problems $\min_{\boldsymbol{\theta}} L_n(\boldsymbol{\theta})$ and $\min_{\boldsymbol{\theta}} R_n(\boldsymbol{\theta}; f)$. This means that the neural network $f(\cdot; \boldsymbol{\theta}^*)$ simultaneously achieves the globally minimal loss value and misclassification rate on the dataset $\mathcal{D}$.

**Theorem 2** *Suppose that Assumption 1 and 2 hold. Suppose that the activation function $\sigma$ is differentiable. Assume that $\tilde{\boldsymbol{\theta}}^* = (\boldsymbol{\theta}^*, \boldsymbol{\theta}_{\exp}^*)$ is a local minimum of the empirical loss function $\tilde{L}_n(\tilde{\boldsymbol{\theta}})$, then $\tilde{\boldsymbol{\theta}}^*$ is a global minimum of $\tilde{L}_n(\tilde{\boldsymbol{\theta}})$. Furthermore, $\boldsymbol{\theta}^*$ achieves the minimum loss value and the minimum misclassification rate on the dataset $\mathcal{D}$, i.e., $\boldsymbol{\theta}^* \in \arg\min_{\boldsymbol{\theta}} L_n(\boldsymbol{\theta})$ and $\boldsymbol{\theta}^* \in \arg\min_{\boldsymbol{\theta}} R_n(\boldsymbol{\theta}; f)$.*

**Remarks**: (i) This theorem is not a direct corollary of the result in the previous section, but the proof ideas are similar. (ii) Due to the assumption on the differentiability of the activation function $\sigma$, Theorem 2 does not apply to the neural networks consisting of non-smooth neurons such as ReLUs, Leaky ReLUs, etc. (iii) Similar to Corollary 1, we will present the following corollary to show that at every local minimum $\tilde{\boldsymbol{\theta}}^* = (\boldsymbol{\theta}^*, \boldsymbol{\theta}_{\exp}^*)$, the neural network $\tilde{f}$ with augmented exponential neurons is equivalent to the original neural network $f$.

**Corollary 2** *Under the conditions in Theorem 2, if $\tilde{\boldsymbol{\theta}}^* = (\boldsymbol{\theta}^*, \boldsymbol{\theta}_{\exp}^*)$ is a local minimum of the empirical loss function $\tilde{L}_n(\tilde{\boldsymbol{\theta}})$, then two neural networks $f(\cdot; \boldsymbol{\theta}^*)$ and $\tilde{f}(\cdot; \tilde{\boldsymbol{\theta}}^*)$ are equivalent, i.e., $f(x; \boldsymbol{\theta}^*) = \tilde{f}(x; \tilde{\boldsymbol{\theta}}^*), \forall x \in \mathbb{R}^d$.*

Corollary 2 further shows that even if we add an exponential neuron to each layer of the original network $f$, at every local minimum of the empirical loss, all exponential neurons are inactive.

## 4.2 Neurons

In this subsection, we will show that even if the exponential neuron is replaced by a monomial neuron, the main result still holds under additional assumptions. Similar to the case where exponential neurons are used, given a neural network $f(x; \boldsymbol{\theta})$, we define a new neural network $\tilde{f}$ by adding the output of a monomial neuron of degree $p$ to the output of the original model $f$, i.e.,

$$\tilde{f}(x; \tilde{\boldsymbol{\theta}}) = f(x; \boldsymbol{\theta}) + a \left(\boldsymbol{w}^\top x + b\right)^p. \tag{7}$$

In addition, the empirical loss function $\tilde{L}_n$ is exactly the same as the loss function defined by Equation (4). Next, we will present the following theorem to show that if all samples in the dataset $\mathcal{D}$ can be correctly classified by a polynomial of degree $t$ and the degree of the augmented monomial is not smaller than $t$ (i.e., $p \geq t$), then every local minimum of the empirical loss function $\tilde{L}_n(\tilde{\boldsymbol{\theta}})$ is also a global minimum. We note that the degree of a monomial is the sum of powers of all variables in this monomial and the degree of a polynomial is the maximum degree of its monomial.

**Proposition 1** *Suppose that Assumptions 1 and 2 hold. Assume that all samples in the dataset $\mathcal{D}$ can be correctly classified by a polynomial of degree $t$ and $p \geq t$. Assume that $\tilde{\boldsymbol{\theta}}^* = (\boldsymbol{\theta}^*, a^*, \boldsymbol{w}^*, b^*)$ is a local minimum of the empirical loss function $\tilde{L}_n(\tilde{\boldsymbol{\theta}})$, then $\tilde{\boldsymbol{\theta}}^*$ is a global minimum of $\tilde{L}_n(\tilde{\boldsymbol{\theta}})$. Furthermore, $\boldsymbol{\theta}^*$ is a global minimizer of both problems $\min_{\boldsymbol{\theta}} L_n(\boldsymbol{\theta})$ and $\min R_n(\boldsymbol{\theta}; f)$.*

**Remarks:** (i) We note that, similar to Theorem 1, Proposition 1 applies to all neural architectures and all neural activation functions defined on $\mathbb{R}$, as we do not require the explicit form of the neural network $f$. (ii) It follows from the Lagrangian interpolating polynomial and Assumption 2 that for a dataset consisted of $n$ different samples, there always exists a polynomial $P$ of degree smaller $n$ such that the polynomial $P$ can correctly classify all points in the dataset. This indicates that Proposition 1 always holds if $p \geq n$. (iii) Similar to Corollary 1 and 2, we can show that at every local minimum $\tilde{\boldsymbol{\theta}}^* = (\boldsymbol{\theta}^*, a^*, \boldsymbol{w}^*, b^*)$, the neural network $\tilde{f}$ with an augmented monomial neuron is equivalent to the original neural network $f$.

## 4.3 Allowing Random Labels

In previous subsections, we assume the realizability of the dataset by the neural network which implies that the label of a given feature vector is unique. It does not cover the case where the dataset contains two samples with the same feature vector but with different labels (for example, the same image can be labeled differently by two different people). Clearly, in this case, no model can correctly classify all samples in this dataset. Another simple example of this case is the mixture of two Gaussians where the data samples are drawn from each of the two Gaussian distributions with certain probability.

In this subsection, we will show that under this broader setting that one feature vector may correspond to two different labels, with a slightly stronger assumption on the convexity of the loss $\ell$, the same result still holds. The formal statement is present by the following proposition.

**Proposition 2** *Suppose that Assumption 1 holds and the loss function $\ell$ is convex. Assume that $\tilde{\boldsymbol{\theta}}^* = (\boldsymbol{\theta}^*, a^*, \boldsymbol{w}^*, b^*)$ is a local minimum of the empirical loss function $\tilde{L}_n(\tilde{\boldsymbol{\theta}})$, then $\tilde{\boldsymbol{\theta}}^*$ is a global minimum of $\tilde{L}_n(\tilde{\boldsymbol{\theta}})$. Furthermore, $\boldsymbol{\theta}^*$ achieves the minimum loss value and the minimum misclassification rate on the dataset $\mathcal{D}$, i.e., $\boldsymbol{\theta}^* \in \arg\min_{\boldsymbol{\theta}} L_n(\boldsymbol{\theta})$ and $\boldsymbol{\theta}^* \in \arg\min_{\boldsymbol{\theta}} R_n(\boldsymbol{\theta}; f)$.*

**Remark:** The differences of Proposition 2 and Theorem 1 can be understood in the following ways. First, as stated previously, Proposition 2 allows a feature vector to have two different labels, but Theorem 1 does not. Second, the minimum misclassification rate under the conditions in Theorem 1 must be zero, while in Proposition 2, the minimum misclassification rate can be nonzero.

## 4.4 High-order Stationary Points

In this subsection, we characterize the high-order stationary points of the empirical loss $\tilde{L}_n$ shown in Section 3.2. We first introduce the definition of the high-order stationary point and next show that every stationary point of the loss $\tilde{L}_n$ with a sufficiently high order is also a global minimum.

**Definition 1** (**$k$-th order stationary point**) *A critical point $\boldsymbol{\theta}_0$ of a function $L(\boldsymbol{\theta})$ is a $k$-th order stationary point, if there exists positive constant $C, \varepsilon > 0$ such that for every $\boldsymbol{\theta}$ with $\|\boldsymbol{\theta} - \boldsymbol{\theta}_0\|_2 \leq \varepsilon$, $L(\boldsymbol{\theta}) \geq L(\boldsymbol{\theta}_0) - C\|\boldsymbol{\theta} - \boldsymbol{\theta}_0\|_2^{k+1}$.*

Next, we will show that if a polynomial of degree $p$ can correctly classify all points in the dataset, then every stationary point of the order at least $2p$ is a global minimum and the set of parameters corresponding to this stationary point achieves the minimum training error.

**Proposition 3** *Suppose that Assumptions 1 and 2 hold. Assume that all samples in the dataset can be correctly classified by a polynomial of degree $p$. Assume that $\tilde{\boldsymbol{\theta}}^* = (\boldsymbol{\theta}^*, a^*, \boldsymbol{w}^*, b^*)$ is a $k$-th order stationary point of the empirical loss function $\tilde{L}_n(\tilde{\boldsymbol{\theta}})$ and $k \geq 2p$, then $\tilde{\boldsymbol{\theta}}^*$ is a global minimum of $\tilde{L}_n(\tilde{\boldsymbol{\theta}})$. Furthermore, the neural network $f(\cdot; \boldsymbol{\theta}^*)$ achieves the minimum misclassification rate on the dataset $\mathcal{D}$, i.e., $\boldsymbol{\theta}^* \in \arg\min_{\boldsymbol{\theta}} R_n(\boldsymbol{\theta}; f)$.*

One implication of Proposition 3 is that if a dataset is linearly separable, then every second order stationary point of the empirical loss function is a global minimum and, at this stationary point, the neural network achieves zero training error. When the dataset is not linearly separable, our result only covers fourth or higher order stationary point of the empirical loss.

## 5 Proof Idea

In this section, we provide overviews of the proof of Theorem 1.

### 5.1 Important Lemmas

In this subsection, we present two important lemmas where the proof of Theorem 1 is based.

**Lemma 1** *Under Assumption 1 and $\lambda > 0$, if $\tilde{\boldsymbol{\theta}}^* = (\boldsymbol{\theta}^*, a^*, \boldsymbol{w}^*, b^*)$ is a local minimum of $\tilde{L}_n$, then (i) $a^* = 0$, (ii) for any integer $p \geq 0$, the following equation holds for all unit vector $\boldsymbol{u} : \|\boldsymbol{u}\|_2 = 1$,*

$$\sum_{i=1}^n \ell'\left(-y_i f(x_i; \boldsymbol{\theta}^*)\right) y_i e^{\boldsymbol{w}^{*\top} x_i + b^*} (\boldsymbol{u}^\top x_i)^p = 0. \tag{8}$$

**Lemma 2** *For any integer $k \geq 0$ and any sequence $\{c_i\}_{i=1}^n$, if $\sum_{i=1}^n c_i(\boldsymbol{u}^\top x_i)^k = 0$ holds for all unit vector $\boldsymbol{u} : \|\boldsymbol{u}\|_2 = 1$, then the $k$-th order tensor $\boldsymbol{T}_k = \sum_{i=1}^n c_i x_i^{\otimes k}$ is a $k$-th order zero tensor.*

### 5.2 Proof Sketch of Lemma 1

**Proof sketch of Lemma 1**($i$): To prove $a^* = 0$, we only need to check the first order conditions of local minima. By assumption that $\tilde{\boldsymbol{\theta}}^* = (\boldsymbol{\theta}^*, a^*, \boldsymbol{w}^*, b^*)$ is a local minimum of $\tilde{L}_n$, then the derivative of $\tilde{L}_n$ with respect to $a$ and $b$ at the point $\tilde{\boldsymbol{\theta}}^*$ are all zeros, i.e.,

$$\nabla_a \tilde{L}_n(\tilde{\boldsymbol{\theta}})\Big|_{\tilde{\boldsymbol{\theta}}=\tilde{\boldsymbol{\theta}}^*} = -\sum_{i=1}^n \ell'\left(-y_i f(x_i; \boldsymbol{\theta}^*) - y_i a^* e^{\boldsymbol{w}^{*\top} x_i + b^*}\right) y_i \exp(\boldsymbol{w}^{*\top} x_i + b^*) + \lambda a^* = 0,$$

$$\nabla_b \tilde{L}_n(\tilde{\boldsymbol{\theta}})\Big|_{\tilde{\boldsymbol{\theta}}=\tilde{\boldsymbol{\theta}}^*} = -a^* \sum_{i=1}^n \ell'\left(-y_i f(x_i; \boldsymbol{\theta}^*) - y_i a^* e^{\boldsymbol{w}^{*\top} x_i + b^*}\right) y_i \exp(\boldsymbol{w}^{*\top} x_i + b^*) = 0.$$

From the above equations, it is not difficult to see that $a^*$ satisfies $\lambda a^{*2} = 0$ or, equivalently, $a^* = 0$. We note that the main observation we are using here is that the derivative of the exponential neuron is itself. Therefore, it is not difficult to see that the same proof holds for all neuron activation function $\sigma$ satisfying $\sigma'(z) = c\sigma(z), \forall z \in \mathbb{R}$ for some constant $c$. In fact, with a small modification of the proof, we can show that the same proof works for all neuron activation functions satisfying $\sigma(z) = (c_1 z + c_0)\sigma'(z), \forall z \in \mathbb{R}$ for some constants $c_0$ and $c_1$. This further indicates that the same proof holds for the monomial neurons and thus the proof of Proposition 1 follows directly from the proof of Theorem 1.

**Proof sketch of Lemma 1**($ii$): The main idea of the proof is to use the high order information of the local minimum to derive Equation (8). Due to the assumption that $\tilde{\boldsymbol{\theta}} = (\boldsymbol{\theta}^*, a^*, \boldsymbol{w}^*, b^*)$ is a local minimum of the empirical loss function $\tilde{L}_n$, there exists a bounded local region such

that the parameters $\tilde{\boldsymbol{\theta}}^*$ achieve the minimum loss value in this region, i.e., $\exists \delta \in (0, 1)$ such that $\tilde{L}_n(\tilde{\boldsymbol{\theta}}^* + \boldsymbol{\Delta}) \geq \tilde{L}_n(\tilde{\boldsymbol{\theta}}^*)$ for $\forall \boldsymbol{\Delta} : \|\boldsymbol{\Delta}\|_2 \leq \delta$.

Now, we use $\delta_a$, $\boldsymbol{\delta_w}$ to denote the perturbations on the parameters $a$ and $\boldsymbol{w}$, respectively. Next, we consider the loss value at the point $\tilde{\boldsymbol{\theta}}^* + \boldsymbol{\Delta} = (\boldsymbol{\theta}^*, a^* + \delta_a, \boldsymbol{w}^* + \boldsymbol{\delta_w}, b^*)$, where we set $|\delta_a| = e^{-1/\varepsilon}$ and $\boldsymbol{\delta_w} = \varepsilon \boldsymbol{u}$ for an arbitrary unit vector $\boldsymbol{u} : \|\boldsymbol{u}\|_2 = 1$. Therefore, as $\varepsilon$ goes to zero, the perturbation magnitude $\|\boldsymbol{\Delta}\|_2$ also goes to zero and this indicates that there exists an $\varepsilon_0 \in (0, 1)$ such that $\tilde{L}_n(\tilde{\boldsymbol{\theta}}^* + \boldsymbol{\Delta}) \geq \tilde{L}_n(\tilde{\boldsymbol{\theta}}^*)$ for $\forall \varepsilon \in [0, \varepsilon_0)$. By the result $a^* = 0$, shown in Lemma 1(i), the output of the model $\tilde{f}$ under parameters $\tilde{\boldsymbol{\theta}}^* + \boldsymbol{\Delta}$ can be expressed by

$$\tilde{f}(x; \tilde{\boldsymbol{\theta}}^* + \boldsymbol{\Delta}) = f(x; \boldsymbol{\theta}^*) + \delta_a \exp(\boldsymbol{\delta_w}^\top x) \exp(\boldsymbol{w}^{*\top} x + b^*).$$

For simplicity of notation, let $g(x; \tilde{\boldsymbol{\theta}}^*, \boldsymbol{\delta_w}) = \exp(\boldsymbol{\delta_w}^\top x) \exp(\boldsymbol{w}^{*\top} x + b^*)$. From the second order Taylor expansion with Lagrangian remainder and the assumption that $\ell$ is twice differentiable, it follows that there exists a constant $C(\tilde{\boldsymbol{\theta}}^*, \mathcal{D})$ depending only on the local minimizer $\tilde{\boldsymbol{\theta}}$ and the dataset $\mathcal{D}$ such that the following inequality holds for every sample in the dataset and every $\varepsilon \in [0, \varepsilon_0)$,

$$\ell(-y_i \tilde{f}(x_i; \tilde{\boldsymbol{\theta}}^* + \boldsymbol{\Delta})) \leq \ell(-y_i f(x_i; \boldsymbol{\theta}^*)) + \ell'(-y_i f(x_i; \boldsymbol{\theta}^*))(-y_i)\delta_a g(x_i; \tilde{\boldsymbol{\theta}}^*, \boldsymbol{\delta_w}) + C(\tilde{\boldsymbol{\theta}}^*, \mathcal{D})\delta_a^2.$$

Summing the above inequality over all samples in the dataset and recalling that $\tilde{L}_n(\tilde{\boldsymbol{\theta}}^* + \boldsymbol{\Delta}) \geq \tilde{L}_n(\tilde{\boldsymbol{\theta}}^*)$ holds for all $\varepsilon \in [0, \varepsilon_0)$, we obtain

$$-\mathrm{sgn}(\delta_a) \sum_{i=1}^{n} \ell'(-y_i f(x_i; \boldsymbol{\theta}^*)) y_i \exp(\varepsilon \boldsymbol{u}^\top x_i) \exp(\boldsymbol{w}^{*\top} x_i + b^*) + [nC(\tilde{\boldsymbol{\theta}}^*, \mathcal{D}) + \lambda/2] \exp(-1/\varepsilon) \geq 0.$$

Finally, we complete the proof by induction. Specifically, for the base hypothesis where $p = 0$, we can take the limit on the both sides of the above inequality as $\varepsilon \to 0$, using the property that $\delta_a$ can be either positive or negative and thus establish the base case where $p = 0$. For the higher order case, we can first assume that Equation (8) holds for $p = 0, ..., k$ and then subtract these equations from the above inequality. After taking the limit on the both sides of the inequality as $\varepsilon \to 0$, we can prove that Equation (8) holds for $p = k + 1$. Therefore, by induction, we can prove that Equation (8) holds for any non-negative integer $p$.

### 5.3 Proof Sketch of Lemma 2

The proof of Lemma 2 follows directly from the results in reference (Zhang et al., 2012). It is easy to check that, for every sequence $\{c_i\}_{i=1}^{n}$ and every non-negative integer $k \geq 0$, the $k$-th order tensor $\boldsymbol{T}_k = \sum_{i=1}^{n} c_i x_i^{\otimes k}$ is a symmetric tensor. From Theorem 1 in (Zhang et al., 2012), it directly follows that

$$\max_{\boldsymbol{u_1}, ..., \boldsymbol{u_k}: \|\boldsymbol{u_1}\|_2 = ... = \|\boldsymbol{u_k}\|_2 = 1} |\boldsymbol{T}_k(\boldsymbol{u_1}, ..., \boldsymbol{u_k})| = \max_{\boldsymbol{u}: \|\boldsymbol{u}\|_2 = 1} |\boldsymbol{T}_k(\boldsymbol{u}, ..., \boldsymbol{u})|.$$

Furthermore, by assumption that $\boldsymbol{T}_k(\boldsymbol{u}, ..., \boldsymbol{u}) = \sum_{i=1}^{n} c_i(\boldsymbol{u}^\top x_i)^k = 0$ holds for all $\|\boldsymbol{u}\|_2 = 1$, then

$$\max_{\boldsymbol{u_1}, ..., \boldsymbol{u_k}: \|\boldsymbol{u_1}\|_2 = ... = \|\boldsymbol{u_k}\|_2 = 1} |\boldsymbol{T}_k(\boldsymbol{u_1}, ..., \boldsymbol{u_k})| = 0,$$

and this is equivalent to $\boldsymbol{T}_k = \boldsymbol{0}_d^{\otimes k}$, where $\boldsymbol{0}_d$ is the zero vector in the $d$-dimensional space.

### 5.4 Proof Sketch of Theorem 1

For every dataset $\mathcal{D}$ satisfying Assumption 2, by the Lagrangian interpolating polynomial, there always exists a polynomial $P(x) = \sum_j c_j \pi_j(x)$ defined on $\mathbb{R}^d$ such that it can correctly classify all samples in the dataset with margin at least one, i.e., $y_i P(x_i) \geq 1, \forall i \in [n]$, where $\pi_j$ denotes the $j$-th monomial in the polynomial $P(x)$. Therefore, from Lemma 1 and 2, it follows that

$$\sum_{i=1}^{n} \ell'(-y_i f(x_i; \boldsymbol{\theta}^*)) e^{\boldsymbol{w}^{*\top} x_i + b^*} y_i P(x_i) = \sum_j c_j \sum_{i=1}^{n} \ell'(-y_i f(x_i; \boldsymbol{\theta}^*)) y_i e^{\boldsymbol{w}^{*\top} x_i + b^*} \pi_j(x_i) = 0.$$

Since $y_i P(x_i) \geq 1$ and $e^{\boldsymbol{w}^{*\top} x_i + b^*} > 0$ hold for $\forall i \in [n]$ and the loss function $\ell$ is a non-decreasing function, i.e., $\ell'(z) \geq 0, \forall z \in \mathbb{R}$, then $\ell'(-y_i f(x_i; \boldsymbol{\theta}^*)) = 0$ holds for all $i \in [n]$. In addition, from the assumption that every critical point of the loss function $\ell$ is a global minimum, it follows that $z_i = -y_i f(x_i; \boldsymbol{\theta}^*)$ achieves the global minimum of the loss function $\ell$ and this further indicates that

$\boldsymbol{\theta}^*$ is a global minimum of the empirical loss $L_n(\boldsymbol{\theta})$. Furthermore, since at every local minimum, the exponential neuron is inactive, $a^* = 0$, then the set of parameters $\tilde{\boldsymbol{\theta}}^*$ is a global minimum of the loss function $\tilde{L}_n(\tilde{\boldsymbol{\theta}})$. Finally, since every critical point of the loss function $\ell(z)$ satisfies $z < 0$, then for every sample, $\ell'(-y_i f(x_i; \boldsymbol{\theta}^*)) = 0$ indicates that $y_i f(x_i; \boldsymbol{\theta}^*) > 0$, or, equivalently, $y_i = \mathrm{sgn}(f(x_i; \boldsymbol{\theta}^*))$. Therefore, the set of parameters $\boldsymbol{\theta}^*$ also minimizes the training error. In summary, the set of parameters $\tilde{\boldsymbol{\theta}}^* = (\boldsymbol{\theta}^*, a^*, \boldsymbol{w}^*, b^*)$ minimizes the loss function $\tilde{L}_n(\tilde{\boldsymbol{\theta}})$ and the set of parameters $\boldsymbol{\theta}^*$ simultaneously minimizes the empirical loss function $L_n(\boldsymbol{\theta})$ and the training error $R_n(\boldsymbol{\theta}; f)$.

## 6 Conclusions and Discussions

One of the difficulties in analyzing neural networks is the non-convexity of the loss functions which allows the existence of many spurious minima with large loss values. In this paper, we prove that for any neural network, by adding a special neuron and an associated regularizer, the new loss function has no spurious local minimum. In addition, we prove that, at every local minimum of this new loss function, the exponential neuron is inactive and this means that the augmented neuron and regularizer improve the landscape of the loss surface without affecting the representing power of the original neural network. We also extend the main result in a few ways. First, while adding a special neuron makes the network different from a classical neural network architecture, the same result also holds for a standard fully connected network with one special neuron added to each layer. Second, the same result holds if we change the exponential neuron to a polynomial neuron with a degree dependent on the data. Third, the same result holds even if one feature vector corresponds to both labels.

This paper is an effort in designing neural networks that are "good". Here "good" can mean various things such as nice landscape, stronger representation power or better generalization, and in this paper we focus on the landscape –in particular, the very specific property "every local minimum is a global minimum". While our results enhance the understanding of the landscape, the practical implications are not straightforward to see since we did not consider other aspects such as algorithms and generalization. It is an interesting direction to improve the landscape results by considering other aspects, such as studying when a specific algorithm will converge to local minima and thus global minima.

## 7 Acknowledgment

Research is supported by the following grants: USDA/NSF CPS grant AG 2018-67007-2837, NSF NeTS 1718203, NSF CPS ECCS 1739189, DTRA Grant DTRA grant HDTRA1-15-1-0003, NSF CCF 1755847 and a start-up grant from Dept. of ISE, University of Illinois Urbana-Champaign.

## Footnotes

*Correpondence to R. Srikant, rsrikant@illinois.edu and Ruoyu Sun, ruoyus@illinois.edu

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
