[Reviews · NeurIPS 2018]

Reviewer 1



STRENGTHS: Overall I found this to be a very interesting and well-written paper. The main contribution is to show that for relatively general network architectures, if either an additional augmenting skip neuron is added directly from the input to the output (or one augmenting neuron is added per layer) all local minima will be globally optimal. In addition, the authors also show that the output of the augmenting neuron(s) will be 0 at all local minima, which implies that one can add an augmenting neuron to a relatively arbitrary network and still be guaranteed that a recovered local minima will also be a global minimum of the original network. The authors provide a comprehensive review of existing results and clearly place their contribution within that context. The proofs are well explained and relatively easy to follow. WEAKNESSES: I found this to be a high quality paper and do not have too many strong criticisms of this paper, but a few points that could improve the paper follow below. 1) There is no experimental testing using the proposed augmenting neurons. A clear prediction of the theory is that the optimization landscape becomes nicer with the addition of the augmenting neuron(s). While understandably these results only apply to local minima and not necessarily arbitrary stationary points (modulo the results of Proposition 3), a fairly simple experiment would be to just add the augmenting neuron and test if the recovered values of the original loss function are smaller when the augmenting neuron(s) are present. 2) Similar to the above comment, a small discussion (perhaps following Proposition 3) regarding the limitations of the results in practical training could be beneficial to some readers (for example, pointing out that most commonly used algorithms are only guaranteed to converge to first order stationary points and not local minima in the general case).

Reviewer 2



=== added after author response === I have two more comments: (a) the scaling issue mentioned by R5 actually leads to an immediate trivial proof for Lemma 1 (i) and invites the following question: the exponential neuron, from an optimization perspective, is entirely "redundant" as it must vanish at any local minimizer but yet it changes the potential set of local minima, by putting more stringent conditions on the local minima. This phenomenon is a bit curious and perhaps deserves more elaboration. (b) I want to emphasize again "eliminating local minima" by itself is no big deal, because you can get a reformulation that eliminates all local-but-not-global minima and yet is NP-hard to solve (e.g., finding a local minimum). This, I am afraid, is likely what is going on here (if you drop the separable assumption). Prove me wrong. === end === Deep neural nets are known to be empirically "immune" to poor local minima, and a lot of recent efforts have been spent on understanding why. The main contribution of this work is to prove that by adding a single exponential function (directly) from input to output and adding a mild l_2 regularizer, the slightly modified, highly nonconvex loss function does not have any non-global local minima. Moreover, all of these local minima actually correspond to the global minima of the original, unmodified nonconvex loss. This surprising result, to the best of my knowledge, is new and of genuine interest. The paper is also very well-written and I enjoyed most in reading this paper out of my 6 assignments. As usual, there is perhaps still some room to improve here. While the main result does appear to be quite intricating at first sight: any local minima is global? and they correspond to the global minima of the original network? Wow! But if we think a bit harder, this result is perhaps not too surprising after all: simply take the Fenchel bi-conjugate of the original loss, then immediately we can conclude any local minima of the biconjugate is global, and under mild assumptions we can also show these local minima correspond to the global minima of the original loss. So, this conclusion itself is not surprising. The nice part of the authors' construction lies in that the modified function is explicitly available and resembles the original neural network so one can actually optimize it for real, while the biconjugate is more of a conceptual tool that is hardly implementable. Nevertheless, I wish the authors had included this comment. It would be a good idea to point out that the so-claimed local minima of the modified loss does exist, for one need only take a global minimizer of the original loss (whose existence we are willing to assume) and augment with 0 to get a local minima of the modified loss. But most importantly, the authors dodged an utterly important question: how difficult it is to find such local minima (of the modified loss)? This question must be answered if we want to actually exploit the nice results that the authors have obtained. My worry is that we probably cannot find good algorithms converging in reasonable amount of time to any of those local minima: simply take a neural network that we know is NP-hard to train, then it follows the modified loss is also NP-hard to train (without the realizability assumption of course). If this question is not answered, I am afraid the author's nice construction would not be too much different from the conceptual biconjugate... Another concern is the authors only focused on the training error, and did not investigate the generalization of the modified network at all.. If we are willing to assume the training data is separable (linear or polynomial), then achieving a zero training error in polytime is really not a big deal (there are plenty of ways). Proposition 2 alleviates some of this concern, but I would suggest the authors add more discussion on the realizability assumption, especially from a non neural network perspective. Some minor comments: Line 77: b_L should be b_{L+1}. Eq (4): the regularization constant lambda can be any positive number? This to me is another alarm: the modified loss likely to be very ill-behaved...

Reviewer 3



This paper considers neural networks and claim that adding one neuron results in making all local minima global for binary classification problem. I might be misinterpreting the theoretical statements of the paper, but I don't quite get why adding the neuron is useful. Assumption 1 readily provides a huge information on the loss function (e.g., every critical point is a global minima) and Assumption 2 implies the the neural net can solve the problem to zero error, both of which (in my opinion) are really strong assumptions. Furthermore, Theorem 1 claims that \tilde{\theta} minimizes \tilde{L} and \delta minimizes L, so I don't quite get why do authors add the extra neuron. They discuss this issue after Corollary 1, but it is not satisfactory. If there exists a local minima that is not global (which is the case in RELU nets as the authors state), then the statement of Theorem 1 doesn't hold, which suggests Assumption 1 is not valid for those scenarios.

Reviewer 4



[Updates] After reading the author's response, I think my concerns about the existence of minima has been partly, but not completely addressed. In the begging of the response a proof for the existence of a global minima is provided. However, an important distinction has been ignored. The author wrote that "By assumption 2, there exists θ* such that f(·; θ*) achieves zero training error". However, the existence a parameter for which all the data can be correctly classified (Assumption 2) is not the same as having a parameter for which the loss function is zero. That is precisely how the counterexample I provided works. Of course, the author could avoid this problem by modifying Assumption 2 to "there exists a parameter for which the loss function is zero", or by adding one assumption stating that $f$ can be rescaled as they did in the response, which I believe they should. Another thing I'm very interested in is how difficult is to find a local minima of the modified network. If I understand correctly, after adding the neuron, each stationary point in the previous network becomes a corresponding saddle point (if one just keep the added neuron inactive) in the modified network (except for the global minima). How does such a loss landscape affect the optimization process? Is it computationally efficient to actually find the minima? How well does the minima generalize? It would be more convincing if the authors can provide some numerical experiments. Overall I believe this is a very good paper, and should be accepted. I've changed my overall score to 7. [Summary of the paper] This paper presents a rather surprising theoretical result for the loss surface of binary classification models. It is shown that under mild assumptions, if one adds a specific type of neuron with skip connection to a binary classification model, as well as a quadratic regularization term to the loss function, then every local minima on the loss surface is also a global minima. The result is surprising because virtually no assumptions have been made about the classification model itself, other than that the dataset is realizable by the model (namely, there exists a parameter under which the model can classify all samples in the dataset correctly), hence the result is applicable to many models. The paper also provides some extensions to their main result. [Quality] I have concerns about the main result of the paper: -- In section 3.2, the authors add the output of an exponential neuron to the output of the network, namely, the new architecture $\tilde{f}$ is defined by $\tilde{f}(x, \theta) = f(x, \theta) + a \exp (w^T x + b)$ Note that the added term has an invariance in its parameters, namely, if one perform the following transformation: $a^\prime = a / C, b^\prime = b + \log C$ then the model will stay exactly the same, i.e., $a^\prime \exp (w^T x + b^\prime) = a \exp (w^T x + b)$ holds for any input $x$. Now, consider the loss function $\tilde{L}_n(\theta, a/C, w, b + \log C)$. Because there is also a regularization term for $a$, by the invariance argument above we can see that $\tilde{L}_n(\theta, a/C, w, b + \log C)$ decreases monotonically as $C$ increases (assuming $a \neq 0$). But as $C$ increases, $b$ is pushed to infinitely faraway. This argument naturally leads to the following concern: Theorem 1 is stated as "if there is a local minima for $\tilde{L}_n, then it is a global minima", but how does one ensure that $\tilde{L}_n$ actually has a local minima at all? To further illustrate my point, consider the following very simple example I constructed. Suppose $f(x, \theta) = 1$, i.e., the model always output 1. Suppose we only have one sample in the dataset, $(x, y) = (0, 1)$. Note that the realizability assumption (Assumption 2) is satisfied. Let the loss function be $l(z) = \max(0, 2 + z)^3$, so that Assumption 1 is satisfied. Finally let $\lambda = 2$. Now, we have $\tilde{L}_n = \max(0, 1 - a \exp(b))^3 + a^2$ One can immediately see that this function has no local minima. To see this, note that when $a = 0$, we have $\tilde{L}_n = 1$; on the other hand, let $a = t$ for some $t > 0$, and $b = - \log t$, and we have $L_n -> 0$ as $t -> 0$, but this would also make $b -> +\infty$. Hence the function has no global minima, and by Theorem 1 it cannot have any local minima. While this observation does not mean Theorem 1 is wrong (because theorem 1 assumes the existence of a local minima), it does limit the scope of Theorem 1 in the case where local minimas do not exist. [Clarity] The paper is well written and well organized. The proof sketch is also easy to follow. [Originality] To the best of my knowledge, the results presented in the paper are original. [Significance] I really like the results presented in this paper. It is quite surprising that by making very simple modifications of the model, one can eliminate bad local minimas, especially given the fact that little assumptions on the model itself are needed. Despite so, I feel that the significance of the results might be slightly less than it appears: -- As mentioned in the [Quality] part, there are cases where the loss function of the modified model has no local minima at all. In such cases, the theorems in the paper do not apply. It is not clear to me what conditions are needed to guarantee the existence of local minimas. It would be nice if the authors can address this issue. -- The theorems in the paper do not actually make any assumptions on the model $f$ except that there exist parameters with which $f$ can correctly classify all samples in the dataset. While this makes the results very general, this unfortunately also implies that the paper is not really about loss surface of neural networks, but rather a general way to modify the loss surface that can be applied to any model so long as the realizability assumption is satisfied. The results seem to have nothing to do with neural networks, and hence it does not really add anything to our understanding of the loss surface of neural networks. The assumptions made in the paper seem reasonable enough to be satisfied in realistic settings. It would be nice if the authors can present some numerical experiments.